# Association between COVID-19 vaccination, SARS-CoV-2 variants, and post COVID-19 condition: A cross-sectional study

**Saif-El-Din El-Akkad**[iD][1], **Selena Shao**[2], **Karen C. Tran**[1,3], **Hiten Naik**[1], **Naveed Janjua**[4,5], **Hind Sbihi**[4,5], **Christopher Carlsten**[1,5,6,7], **James A. Russell**[7], **Adeera Levin**[1], **Alyson W. Wong**[1,3]*

1 Department of Medicine, University of British Columbia, Vancouver, British Columbia, Canada, 2 BC Renal, Provincial Health Services Authority, Vancouver, British Columbia, Canada, 3 Centre for Advancing Health Outcomes, St. Paul's Hospital, University of British Columbia, Vancouver, British Columbia, Canada, 4 Data and Analytic Services, BC Centre for Disease Control, Vancouver, British Columbia, Canada, 5 School of Population and Public Health, The University of British Columbia, Vancouver, British Columbia, Canada, 6 Centre for Lung Health, Vancouver Coastal Health Research Institute, Vancouver, British Columbia, Canada, 7 Centre for Heart Lung Innovation, St. Paul's Hospital, University of British Columbia, Vancouver, British Columbia, Canada

* awong@providencehealth.bc.ca

## Abstract

### Background

Individuals may experience persistent symptoms after recovering from coronavirus disease 2019 (COVID-19), a condition referred to as post COVID-19 condition (PCC). Patient-reported outcome measures (PROMs) evaluate a patient's health status and can be used to quantify symptom severity from the patient's perspective. The impact of COVID-19 vaccination and variant of infection on PCC is not well understood. We therefore sought to explore vaccination and variant trends among individuals with PCC and investigate their association with abnormal PROMs.

### Methods

Patients seen at Post-COVID-19 Clinics across British Columbia, Canada between March 2020 – Oct 2022 were included in the study. Those who had persistent symptoms, at least one abnormal PROM, and completed a baseline questionnaire within 6 months of infection were included. The following PROMs were used: Fatigue Severity Score, University of California San Diego Shortness of Breath Questionnaire, Post Traumatic Stress Disorder (PTSD) Score, Generalized Anxiety Disorder-2 and Patient Health Questionnaire-2. Vaccination status was categorized based on the number and timing of vaccinations relative to SARS-CoV-2 infection. Logistic regression was used to evaluate the association between COVID-19 vaccination status or SARS-CoV-2 variant and the likelihood of reporting abnormal PROMs.

**Data availability statement:** According to the University of British Columbia Clinical Research Ethics Board (UBC CREB), we are prohibited from sharing data publicly as there is potentially sensitive information. Data requests may be sent to the UBC CREB at the following address: Room 210, Research Pavilion - 828 West 10th Ave, Vancouver, BC Canada V5Z1M9.

**Funding:** AWW received funding from the Canadian Lung Association (CLA) / Canadian Institutes of Health Research (CIHR) Respiratory Health Effects of PCC Grant (Funding reference 181075). The funders had no role in study design, data collection and analysis, decision to publish, or preparation of the manuscript.

**Competing interests:** The authors have declared that no competing interests exist.

## Results

The study included 1,587 participants (mean age 52 ± 15 years, 45% male, 58% vaccinated). In the adjusted models, full vaccination among non-hospitalized patients was associated with a reduced likelihood of reporting PTSD. Hospitalized patients infected with the alpha and delta variants were more likely to report dyspnea, while those infected with the gamma variant were less likely to report PTSD.

## Conclusions

Patients who were partially or fully vaccinated did not have increased risk of reporting common PCC symptoms. Infection with the alpha and delta variants was associated with increased likelihood of reporting dyspnea, which may be related to the severity of acute illness and associated impairments in lung function.

## Introduction

Persistent symptoms following Coronavirus Disease 2019 (COVID-19) is common and is referred to as post COVID-19 condition (PCC) [1]. A meta-analysis of 194 studies and 735,000 patients infected with severe acute respiratory syndrome coronavirus-2 (SARS-CoV-2), who had varying severities of acute illness, found that 45% of patients met diagnostic criteria for PCC [2]. A better understanding of how certain SARS-CoV-2 variants and vaccination status relate to PCC symptoms is needed to inform preventative measures and treatments.

COVID-19 vaccinations reduce disease severity and recent systematic reviews have also suggested that vaccination may have protective effects against PCC [1,3]. However, existing evidence is derived primarily from cross-sectional studies with restricted follow up time [3]. The use of different methods to collect symptom data (e.g., patient reported outcome measures [PROMs] that are standardized and validated versus patient questionnaires) and data sources (e.g., patient surveys versus chart reviews) also limits comparability between studies.

Over the course of the pandemic, SARS-CoV-2 has mutated to produce multiple variants with varying degrees of transmissibility, virulence, and clinical severity [4,5]. The differences in PCC symptoms among variants may be due to varying involvement along the respiratory tract, severity of acute infection, and existing degrees of immunity at the time of infection [6,7]. Understanding whether variants impact PCC symptoms may offer unique insights into the disease's pathophysiologic mechanisms [8].

The impact of vaccination status, vaccination timing, and SARS-CoV-2 variants on the development and severity of PCC symptoms is not well understood. We sought to explore the association between 1) a patient's COVID-19 vaccination status and 2) the SARS-CoV-2 variant at time of acute illness with the presence or absence of abnormal PROMs after recovery.

## Methods

### Study population

We studied a longitudinal cohort of patients seen in one of four Post-COVID-19 recovery clinics (PCRCs) in British Columbia, Canada between March 2020 and October 2022. The data was accessed for research purposes between 25-05-2022 and 19-06-2024. The PCRC consists of a multidisciplinary healthcare team that includes physicians, nurses, and other allied health professionals [9]. Hospitalized patients were automatically referred to the PCRC at the time of discharge, while non-hospitalized patients with COVID-19 could be referred by a physician to the PCRC if they had persistent symptoms ≥4 weeks after their acute COVID-19 illness. Patients were included in the study if they were 1) over the age of 18 years, 2) had a SARS-CoV-2 infection as confirmed via either positive SARS-CoV-2 reverse transcriptase-polymerase chain reaction (RT-PCR) test provided by the British Columbia Centre for Disease Control (BCCDC) or by a COVID-19 positive date listed on the PCRC referral and 3) had PCC as defined by the presence of at least one abnormal PROM on their baseline questionnaire that was completed within 6 months of their acute illness. Ethics approval for this study was obtained from the Research Ethics Board (REB) at the University of British Columbia (H2102660) and the REB waived the need for individual consent as it involved secondary analyses of data collected as part of clinical care.

### Measurements

Patient characteristics and features of their acute SARS-CoV-2 infection were collected from baseline questionnaires and chart reviews. The following validated PROMs were included (abnormal scores are shown in parentheses): University of California San Diego (UCSD) shortness of breath questionnaire (≥10/120) [10], Fatigue Severity Scale (FSS) (≥4/7 if mean reported or ≥ 36/63 if sum reported) [11], Generalized Anxiety Disorder-2 (≥3/6) [12], Patient Health Questionnaire-2 (≥3/6) [13], Primary Care Post Traumatic Stress Disorder (PTSD) Screen for The Diagnostic and Statistical Manual of Mental Health Fifth Edition (DSM-5) (≥3/5) [14]. Patients had pulmonary function tests (PFTs) if clinically indicated. PFTs completed within 12 months of baseline questionnaire were included.

### Vaccination status and variants

If a patient had multiple PCR confirmed SARS-CoV-2 infections, the infection that triggered a referral to the PCRC was used for analysis. Vaccination status was categorized as follows: 1) Unvaccinated (individuals who had either never been vaccinated or had received their first dose ≤13 days before their positive SARS-CoV-2 test), 2) Vaccinated after (individuals who were vaccinated after SARS-CoV-2 infection), 3) Partially vaccinated (those who received a single vaccine of the primary vaccination series ≥14 days before, two or more vaccines >165 days before, or at least one dose of the single dose viral vector-based vaccine (Johnson & Johnson) vaccine >165 days before the index date and 4) Fully vaccinated (individuals who had received one single dose vaccine or two or more doses of the primary series vaccines < 165 days before the index date. These categories served to capture both differences in vaccination status at the time of infection and waning immunity following vaccination as there is robust evidence suggesting that vaccine protection against COVID-19 wanes over time [15,16]. Although both the unvaccinated and vaccinated after groups did not have immunity from vaccines at the time of infection, these groups were categorized separately as they may have different perceptions towards vaccination and symptoms after COVID-19. It also enabled exploration on the impact of vaccination after COVID-19 and PCC.

For variant-related analyses, patients were categorized based on the SARS-CoV-2 variant of concern (VOC: alpha, delta, gamma, or omicron) that they were infected with during their acute illness. VOCs circulated in British Columbia during the following approximate time periods: alpha (January – June 2021), gamma (February – July 2021), delta (May 2021 – December 2021) and omicron (December 2021 onwards) [17]. Data were limited to these four dominant VOCs based on data availability and categorization used by the BCCDC Public Health Laboratory (PHL) during the study duration (March 2020 to October 2022). The BCCDC

PHL continuously monitors for VOCs, variants of interest (VOIs), and variants under monitoring (VUMs). Whole genome sequencing (WGS) was primarily used for sample characterization, but the BCCDC also relied on VOC screening by qPCR. The BCCDC PHL optimised its sequencing strategy based on available laboratory capacity and public health needs. For samples that underwent both qPCR single nucleotide polymorphism (SNP) screening and further confirmation by WGS, lineage results from WGS were used.

## Statistical analysis

The cohort was stratified into hospitalized and non-hospitalized groups due to the differences in referral criteria to the PCRC. Logistic regression was used to estimate the associations between 1) vaccination status (unvaccinated, vaccinated after, partially vaccinated, and fully vaccinated) and 2) SARS-CoV-2 variant (alpha, delta, gamma) and the presence of an abnormal PROM. Analysis of the omicron variant was not conducted due to small sample size (n = 51). All analyses were adjusted for age, sex and comorbidities (heart disease, diabetes, asthma, chronic obstructive pulmonary disease, malignancy, or depression). The variant analysis also adjusted for vaccination status (which was used as a surrogate for immunity). The reference groups in the vaccination and variant analyses were participants who were unvaccinated at the time of SARS-CoV-2 infection and those who were infected with the wild-type SARS-CoV-2 strain, respectively.

Post-hoc analysis of pulmonary function tests across variant groups were conducted to understand whether physiologic differences in lung function could account for differences in PROMs. Kruskal-Wallis tests were used to compare differences in percent predicted forced vital capacity (FVC), forced expiratory volume in one second (FEV1), total lung capacity (TLC) and diffusing capacity of lungs for carbon dioxide (DLCO) across variant groups. Statistical analyses were performed using R Version 4.0.3.

# Results

## Patient characteristics

There were 1587 participants included in the study (Fig 1): 1481 (93%) participants had a confirmed positive SARS-CoV-2 PCR test, while 106 (7%) participants had a confirmed SARS-CoV-2 infection date but no documented PCR test upon referral to the PCRC. The mean age for the total cohort was 52 years and similar between the vaccination groups (Table 1). There were more females in the partially (69%) and fully (68%) vaccinated groups compared to the unvaccinated (52%) and vaccinated after (51%) groups. The most common comorbidities in the cohort were depression (33%), hypertension (29%), diabetes (17%) and asthma (17%). Approximately half the cohort was hospitalized for acute COVID-19, with a lower percentage of hospitalizations in the fully vaccinated group (21%) compared to 58% in the unvaccinated and 57% in the vaccinated after groups. PROM scores at the initial clinic visit were also collected and the proportion of participants who reported abnormal scores was calculated (S1 Table).

Among the 126 participants who were partially vaccinated prior to infection, the median time from vaccination to acute COVID-19 illness was 179 days (interquartile range [IQR] 71–197 days); of whom 44 (35%) participants were vaccinated once, 79 (63%) were vaccinated twice and 3 (2%) were vaccinated three times. Among the 178 participants who were fully vaccinated prior to infection, the median time from vaccination to infection was 83 days (IQR 42–124), 2 (1%) were vaccinated once (single dose vaccines), 101 (57%) were vaccinated twice, 73 (41%) were vaccinated three times and 2 (1%) were vaccinated four times. For those vaccinated after infection, the median time from acute COVID-19 illness to vaccination was 85 days (IQR 63–108 days); 344 (63%) patients were vaccinated once, 196 (36%) were vaccinated twice and 4 (1%) were vaccinated three times.

There were 361 participants diagnosed prior to January 2021 and thereby infected with the wild-type strain. There were 773 participants with variant lineage data available and were categorized as follows: 198 (26%) were infected with alpha (B.1.1.7), 250 (32%) with gamma (P.1, P.1.*), 184 (24%) with delta (B.1.617.2, AY.*) and 51 (7%) with omicron (B.1.1.529, BA.*). Ninety (11%) patients had other non-VOC strains that were not included in the analysis (S2 Table).

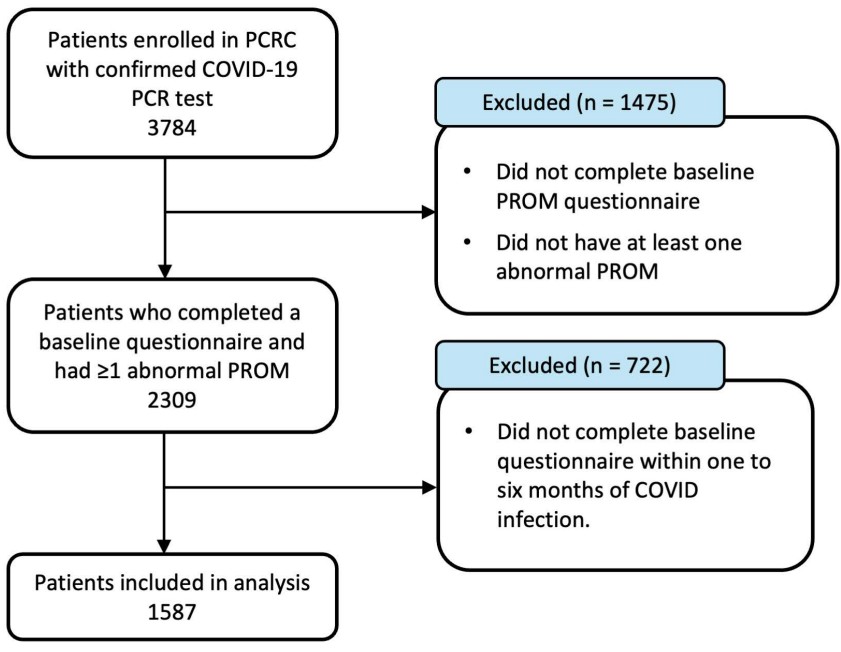

**Fig 1. Flowchart of cohort selection.**

## Vaccination status and abnormal PROMs

In the unadjusted analysis, non-hospitalized patients who were partially vaccinated (odds ratio [OR] 2.51; 95% confidence interval [CI] 1.21–5.90; p-value 0.02) or fully vaccinated (OR 2.10; 95% CI 1.15–4.10; p-value 0.02) before acute SARS-CoV-2 infection were more likely to report fatigue than unvaccinated patients (Table 2). However, after adjusting for age, sex and comorbidities, partial or full vaccination was not associated with any abnormal PROMs. Those who were fully vaccinated were less likely to report PTSD (OR 0.57; 95% CI 0.33–0.97; p-value 0.04) compared to those who were unvaccinated in the non-hospitalized group (Fig 2).

## SARS-CoV-2 variants and abnormal PROMs

In the unadjusted analysis, hospitalized patients infected with the alpha variant (OR 2.60; 95% CI 1.44–4.80; p-value 0.002) and non-hospitalized patients infected with the delta variant (OR 2.70; 95% CI 1.22–6.90; p-value 0.02) were more likely to report dyspnea compared to patients infected with the wildtype strain (Table 3). After adjusting for age, sex, comorbidities and vaccination status, hospitalized patients with the alpha (OR 2.71; 95% CI 1.43–5.30; p-value 0.003) and delta variant (OR 1.84; 95% CI 1.02–3.40; p-value 0.047) were still more likely to report dyspnea (Fig 3). Conversely, hospitalized patients infected with the gamma variant were less likely to report depression in the unadjusted (OR 0.55; 95% CI 0.33–0.91; p-value 0.02) and adjusted (OR 0.56; 95% CI 0.32–0.98; p-value 0.045) analyses.

## Pulmonary function among variants

There were 233 patients with PFT data available for the post-hoc analysis (wild-type n = 152, alpha n = 32, delta n = 11, gamma n = 38) (Table 4). There was a significant difference in the DLCO %-predicted between patients infected with the alpha variant (mean ± standard deviation [SD] 71 ± 17%; p-value < 0.001), delta variant (mean ± SD 70 ± 24%; p-value 0.03) and the gamma variant (mean ± SD 71 ± 20%; p-value < 0.001), compared to those infected with the wild-type strain

**Table 1.** Characteristics of patients with PCC based on vaccination status (n = 1587).

| Characteristic | Total Cohort (n = 1587) | Vaccination Status | | | |
| --- | --- | --- | --- | --- | --- |
| | | Unvaccinated (n = 739) | Partially vaccinated (n = 126) | Fully vaccinated (n = 178) | Vaccinated after (n = 544) |
| Age, mean ± SD | 52 ± 15 | 52 ± 15 | 50 ± 16 | 49 ± 15 | 52 ± 15 |
| Male sex (n, %) | 715 (45) | 354 (48) | 39 (31) | 57 (32) | 265 (49) |
| **Race (n, %)** | | | | | |
| White | 727 (46) | 304 (41) | 85 (68) | 134 (75) | 204 (38) |
| Asian | 526 (33) | 239 (32) | 25 (20) | 22 (12) | 240 (44) |
| Latin | 56 (4) | 31 (4) | 3 (2) | 2 (1) | 20 (4) |
| Indigenous | 52 (3) | 19 (3) | 6 (5) | 8 (4) | 19 (3) |
| Black | 15 (1) | 6 (1) | 0 (0) | 1 (1) | 8 (1) |
| Other | 96 (6) | 46 (6) | 3 (2) | 8 (5) | 39 (7) |
| Unknown | 115 (7) | 94 (13) | 4 (3) | 3 (2) | 14 (3) |
| **Ever Smoker (n, %)** | | | | | |
| Yes | 465 (29) | 211 (29) | 40 (32) | 68 (38) | 146 (27) |
| No | 992 (63) | 417 (56) | 84 (67) | 107 (60) | 384 (70) |
| Unknown | 130 (8) | 111 (15) | 2 (1) | 3 (2) | 14 (3) |
| **Comorbidities (n, %)** | | | | | |
| Heart Disease | 162 (10) | 77 (10) | 11 (9) | 18 (10) | 56 (10) |
| Diabetes | 273 (17) | 123 (17) | 23 (18) | 20 (11) | 107 (20) |
| Hypertension | 463 (29) | 200 (27) | 39 (31) | 52 (29) | 172 (32) |
| Asthma | 276 (17) | 125 (17) | 26 (21) | 44 (25) | 81 (15) |
| COPD | 105 (7) | 42 (6) | 15 (12) | 17 (10) | 31 (6) |
| Malignancy | 48 (3) | 18 (2) | 6 (5) | 8 (4) | 16 (3) |
| Depression | 530 (33) | 216 (29) | 62 (49) | 77 (43) | 175 (32) |
| **Hospitalized (n, %)** | | | | | |
| Yes – Due to COVID | 810 (51) | 429 (58) | 31 (25) | 38 (21) | 312 (57) |
| Yes – Not due to COVID | 65 (4) | 25 (3) | 4 (3) | 6 (4) | 30 (6) |
| No | 712 (45) | 285 (39) | 91 (72) | 134 (75) | 202 (37) |
| Hospitalization (days; median, IQR) | 10 (5-17) | 10 (6-18) | 9 (6-22) | 9 (5-21) | 9 (5-16) |
| **Intensive Care (n, %)** | | | | | |
| Yes | 310 (20) | 171 (23) | 6 (5) | 9 (5) | 124 (23) |
| No | 1,277 (80) | 568 (77) | 120 (95) | 169 (95) | 420 (77) |
| Time from vaccination to infection (days; median, IQR)* | 106 (89-135) | 5 (3-7) | 179 (71-197) | 83 (42-124) | N/A |
| Time from infection to baseline questionnaire (days; median, IQR) | 106 (89-135) | 97 (85-120) | 114 (91-137) | 112 (91-141) | 116 (96-149) |
| **Vaccination type (n, %)*** | | | | | |
| mRNA vaccinated | 861 (54) | 69 (9) | 113 (90) | 152 (85) | 527 (97) |
| Non-mRNA vaccinated | 38 (2) | 7 (1) | 4 (3) | 12 (7) | 15 (3) |
| Combination | 25 (2) | 0 (0) | 9 (7) | 14 (8) | 2 (0) |
| **Reported abnormal PROM (n, %)** | | | | | |
| Fatigue | 1093 (69) | 483 (65) | 102 (81) | 150 (84) | 358 (66) |
| Dyspnea | 1236 (78) | 562 (76) | 109 (87) | 151 (85) | 414 (76) |
| PTSD | 333 (21) | 141 (19) | 30 (24) | 38 (21) | 124 (23) |
| Anxiety | 520 (33) | 229 (31) | 52 (41) | 72 (40) | 167 (31) |

*(Continued)*

**Table 1.** (Continued)

| Characteristic | Total Cohort (n = 1587) | Vaccination Status | | | |
| | | Unvaccinated (n = 739) | Partially vaccinated (n = 126) | Fully vaccinated (n = 178) | Vaccinated after (n = 544) |
|---|---|---|---|---|---|
| Depression | 484 (30) | 205 (28) | 48 (38) | 70 (39) | 161 (30) |

*Abbreviations:* COPD, chronic obstructive pulmonary disease; PTSD post-traumatic stress disorder; PROM, patient reported outcome measure; IQR, interquartile range.

*76 patients had received their first dose ≤13 days before their positive SARS-CoV-2 test and were considered unvaccinated as a result.

**Table 2.** Unadjusted analysis of vaccination status and abnormal PROMs (n = 1587).

| Characteristic | Fatigue | | Dyspnea | | PTSD | | Anxiety | | Depression | |
| | OR (95% CI) | p-value | OR (95% CI) | p-value | OR (95% CI) | p-value | OR (95% CI) | p-value | OR (95% CI) | p-value |
|---|---|---|---|---|---|---|---|---|---|---|
| **Non-Hospitalized (n = 712)** | | | | | | | | | | |
| Partially vaccinated (n = 91) | 2.51 (1.21-5.9) | **0.02** | 1.6 (0.83-3.3) | 0.18 | 0.93 (0.52-1.61) | 0.80 | 1.16 (0.72-1.97) | 0.54 | 1.36 (0.84-2.19) | 0.21 |
| Fully vaccinated (n = 134) | 2.10 (1.15-4.1) | **0.02** | 1.7 (0.95-3.1) | 0.08 | 0.81 (0.49-1.32) | 0.40 | 0.99 (0.65-151) | 0.98 | 1.14 (0.75-1.74) | 0.53 |
| Vaccinated after (n = 202) | 0.96 (0.61-1.5) | 0.86 | 1.2 (0.75-1.9) | 0.46 | 1.02 (0.67-1.55) | 0.93 | 0.94 (0.65-1.36) | 0.75 | 1.06 (0.73-1.54) | 0.74 |
| **Hospitalized (n = 875)** | | | | | | | | | | |
| Partially vaccinated (n = 35) | 1.17 (0.56-2.50) | 0.67 | 2.60 (1.00-8.90) | 0.08 | 1.90 (0.80-4.08) | 0.13 | 1.32 (0.58-2.78) | 0.49 | 0.96 (0.37-2.17) | 0.92 |
| Full protection (n = 44) | 1.50 (0.79-3.00) | 0.23 | 1.14 (0.56-2.50) | 0.73 | 1.50 (0.67-3.02) | 0.31 | 1.57 (0.79-2.99) | 0.18 | 1.77 (0.89-3.39) | 0.09 |
| Vaccinated after (n = 342) | 0.99 (0.75-1.30) | 0.97 | 0.89 (0.65-1.20) | 0.49 | 1.40 (0.99-2.04) | 0.06 | 0.99 (0.71-1.37) | 0.95 | 1.08 (0.77-1.51) | 0.64 |

*Abbreviations:* OR odds ratio; CI, confidence internal; PTSD, post-traumatic stress disorder; PROM, patient reported outcome measure.

Significant values (p < 0.05) are in bold.

Reference groups: Unvaccinated patients: non-hospitalized (n=285) and hospitalized (n=454).

(mean ± SD 86 ± 20%). There was a borderline significant difference in FVC among the variants (p = 0.05), while no differences were seen in the other PFT values.

## Discussion

This study aimed to explore the association between 1) COVID-19 vaccination status and 2) SARS-CoV-2 variants and the presence of PCC symptoms (fatigue, dyspnea, anxiety, depression, PTSD) using PROMs. In the adjusted analyses, non-hospitalized patients who were fully vaccinated prior to their COVID-19 illness were less likely to report PTSD symptoms compared to unvaccinated patients; while hospitalized patients infected with alpha or gamma variants were more likely to report dyspnea and less likely depression, respectively, compared to those infected with the wildtype strain.

Our study shows that vaccination did not increase the likelihood of reporting common PCC symptoms. In fact, fully vaccinated patients were less likely to report PTSD symptoms which is in keeping with other studies [18,19]; however protective trends with the other PCC symptoms were not seen. There are several possible explanations for our findings. A large proportion (35%) of our cohort who were partially vaccinated prior to acute infection received only one vaccination. The protective benefits of vaccination are likely additive [18]. In a meta-analysis, receiving two-doses of vaccine before

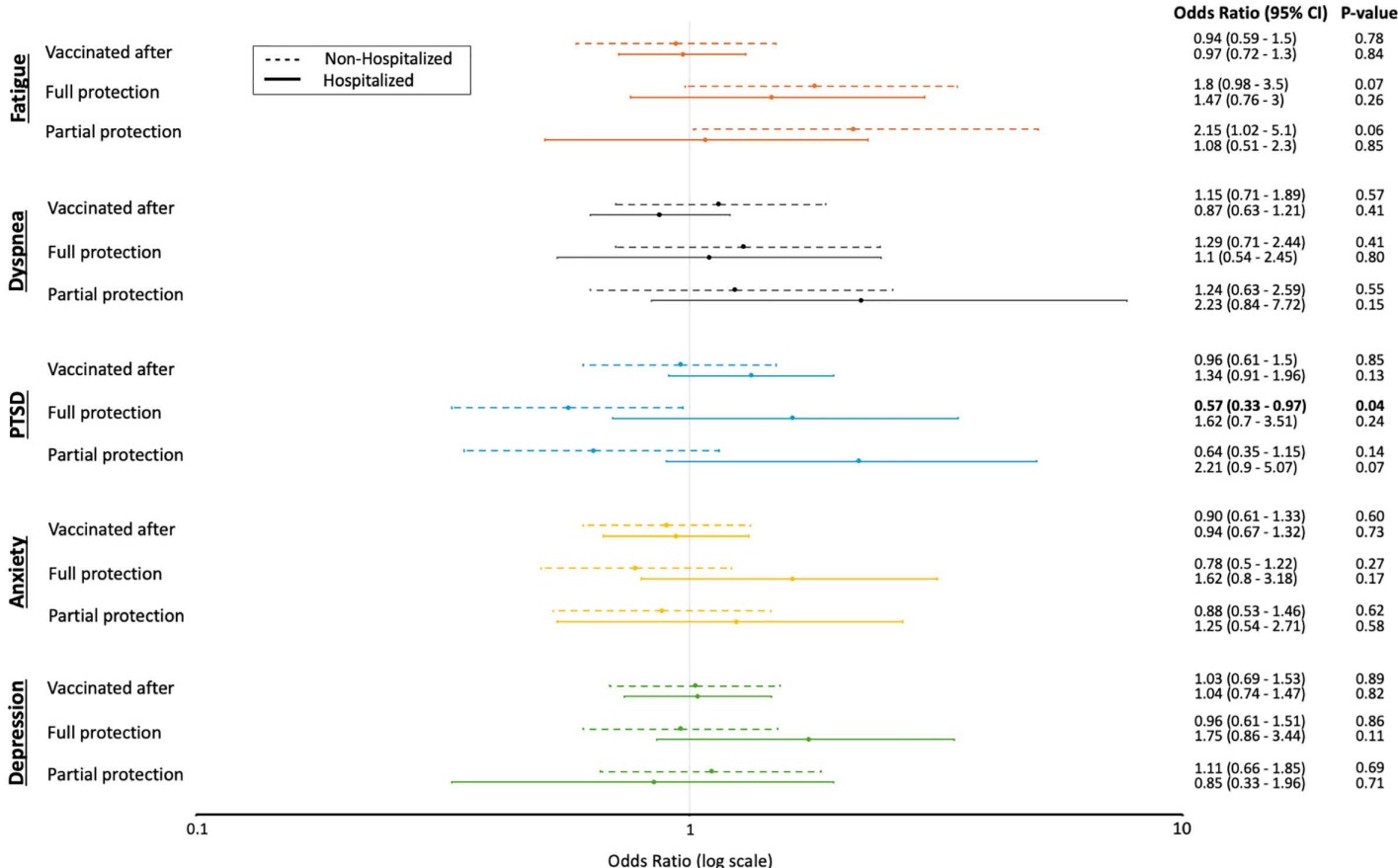

**Fig 2. Association between vaccination status and abnormal PROM adjusted for age, sex, and comorbidities.** Reference group: unvaccinated patients.

COVID was associated with a reduced incidence of PCC compared to no vaccine, but not in patients who received only a single vaccine dose [18]. Second, the perception of symptoms may be different between these groups. Patients who were unvaccinated may be less likely to perceive and report persistent symptoms as abnormal. Importantly, our study demonstrates that amongst those with PCC, immunization either prior to or after acute SARS-CoV-2 infection does not increase the likelihood of common PCC symptoms and can be used to inform discussions about COVID-19 vaccination with patients.

Our study found an association between hospitalized COVID-19 patients infected with the alpha and delta variants and an increased presence of dyspnea compared to the wild-type group, after adjusting for age, sex, comorbidities and vaccination status. The alpha variant was prevalent earlier in the pandemic (January-June 2021) when vaccination rates were low due to lack of vaccine availability [17], while the delta variant was associated with higher viral loads and hospitalization rates [20,21]. Patients infected with both the alpha and delta variants may have experienced more severe illness, which in turn may have led to higher rates of reported dyspnea. To better understand this, we first explored hospitalization trends across wild-type and variant groups; hospitalization rates were proportionally higher amongst participants infected with the alpha (68%) and delta (65%) variants compared to the wild-type (45%) suggesting more severe disease amongst these variant groups. ICU admissions amongst hospitalized patients infected with the delta variant (48%) were also greater than those infected with alpha (34%) and wild-type (37%) strains. We also performed a post-hoc analysis comparing PFT

**Table 3. Unadjusted analysis of SARS-CoV-2 variant and abnormal PROMs (n = 993).**

| Characteristic | Fatigue | | Dyspnea | | PTSD | | Anxiety | | Depression | |
|---|---|---|---|---|---|---|---|---|---|---|
| | OR (95% CI) | p-value | OR (95% CI) | p-value | OR (95% CI) | p-value | OR (95% CI) | p-value | OR (95% CI) | p-value |
| **Non-Hospitalized (n = 374)** | | | | | | | | | | |
| Alpha (n = 64) | 2.10 (0.98-4.70) | 0.07 | 1.30 (0.66-2.60) | 0.48 | 1.47 (0.75-2.80) | 0.26 | 1.00 (0.56-1.80) | 0.99 | 1.75 (0.99-3.11) | 0.05 |
| Delta (n = 65) | 1.80 (0.90-4.10) | 0.11 | 2.70 (1.22-6.90) | **0.02** | 1.71 (0.89-3.23) | 0.10 | 1.42 (0.80-2.50) | 0.23 | 1.55 (0.87-2.75) | 0.14 |
| Gamma (n = 47) | 1.90 (0.85-4.90) | 0.14 | 1.40 (0.64-3.20) | 2.43 | 1.72 (0.82-3.48) | 0.14 | 0.91 (0.47-1.70) | 0.78 | 0.88 (0.43-1.70) | 0.70 |
| **Hospitalized (n = 619)** | | | | | | | | | | |
| Alpha (n = 134) | 1.50 (0.93-2.40) | 0.10 | 2.60 (1.44-4.80) | **0.002** | 1.02 (0.58-1.79) | 0.95 | 1.22 (0.73-2.10) | 0.45 | 1.22 (0.73-2.03) | 0.45 |
| Delta (n = 119) | 1.30 (0.79-2.10) | 0.31 | 1.60 (0.93-2.90) | 0.09 | 1.06 (0.59-1.89) | 0.84 | 0.90 (0.51-1.60) | 0.70 | 0.80 (0.46-1.39) | 0.44 |
| Gamma (n=203) | 1.10 (0.70-1.60) | 0.75 | 1.10 (0.71-1.80) | 0.63 | 0.86 (0.51-1.44) | 0.56 | 0.91 (0.56-1.50) | 0.69 | 0.55 (0.33-0.91) | **0.02** |

*Abbreviations:* OR odds ratio; CI, confidence internal; PTSD, post-traumatic stress disorder; PROM, patient reported outcome measure.

Significant values (p < 0.05) are in bold.

Reference group: non-hospitalized wild-type (n = 198), hospitalized wild-type (n = 163).

measurements across variant groups to evaluate whether there was a physiologic explanation for the increase in dyspnea seen with the alpha and delta variants. The alpha and delta variant groups had significantly lower mean DLCO %-predicted (alpha 71 ± 17%, p-value < 0.001 and delta 70 ± 24%, p-value 0.03) versus the wild-type group (86 ± 20%), which suggests that increased dyspnea in the alpha and delta variant groups could be due to greater lung function abnormalities from parenchymal and/or pulmonary vascular injury during the acute illness. Other studies have also found different symptoms associated with specific SARS-CoV-2 variants [22–24]. We suspect that differences in the types and severity of symptoms among the variants is a complex interplay between biologic causes (e.g., virulence) and psychosocial factors (e.g., impact of anxiety in the general population or public health orders on individuals); however, there could be different pathophysiologic mechanisms associated with SARS-CoV-2 variants that result in certain symptoms and phenotypes after the acute illness. Further studies are required to explore this hypothesis.

Although there are consensus definitions for PCC, there is a lack of standardization in identifying clinically significant PCC symptoms which limits comparability between studies [25,26]. We addressed this key challenge by utilizing PROMs to study PCC. Patients also had to meet validated thresholds for these PROMs which ensured the degree of symptoms reported were clinically meaningful. The use of PROMs reduces heterogeneity within the literature, allows for objective identification and monitoring of clinically relevant PCC symptoms, and identifies future avenues for clinical trials that focus on patient-centered outcomes. In the absence of clear biological markers for PCC severity, PROMs have emerged as a critical outcome measure in the study of PCC and their consistent adoption across studies would support data aggregation and comparison across studies [27–32].

This study is limited by the absence of PROM data prior to SARS-CoV-2 infection and lack of a control cohort, which restricts our ability to attribute the presence of abnormal PROMs to PCC. We attempted to focus in on the relationship between SARS-CoV-2 infection and abnormal PROMs by adjusting for comorbidities to address the impact that pre-existing conditions and impaired health may have on PROMs. We also stratified patients based on whether or not they were hospitalized for COVID-19 as the severity of acute illness may increase the development and severity of PCC [2].

 

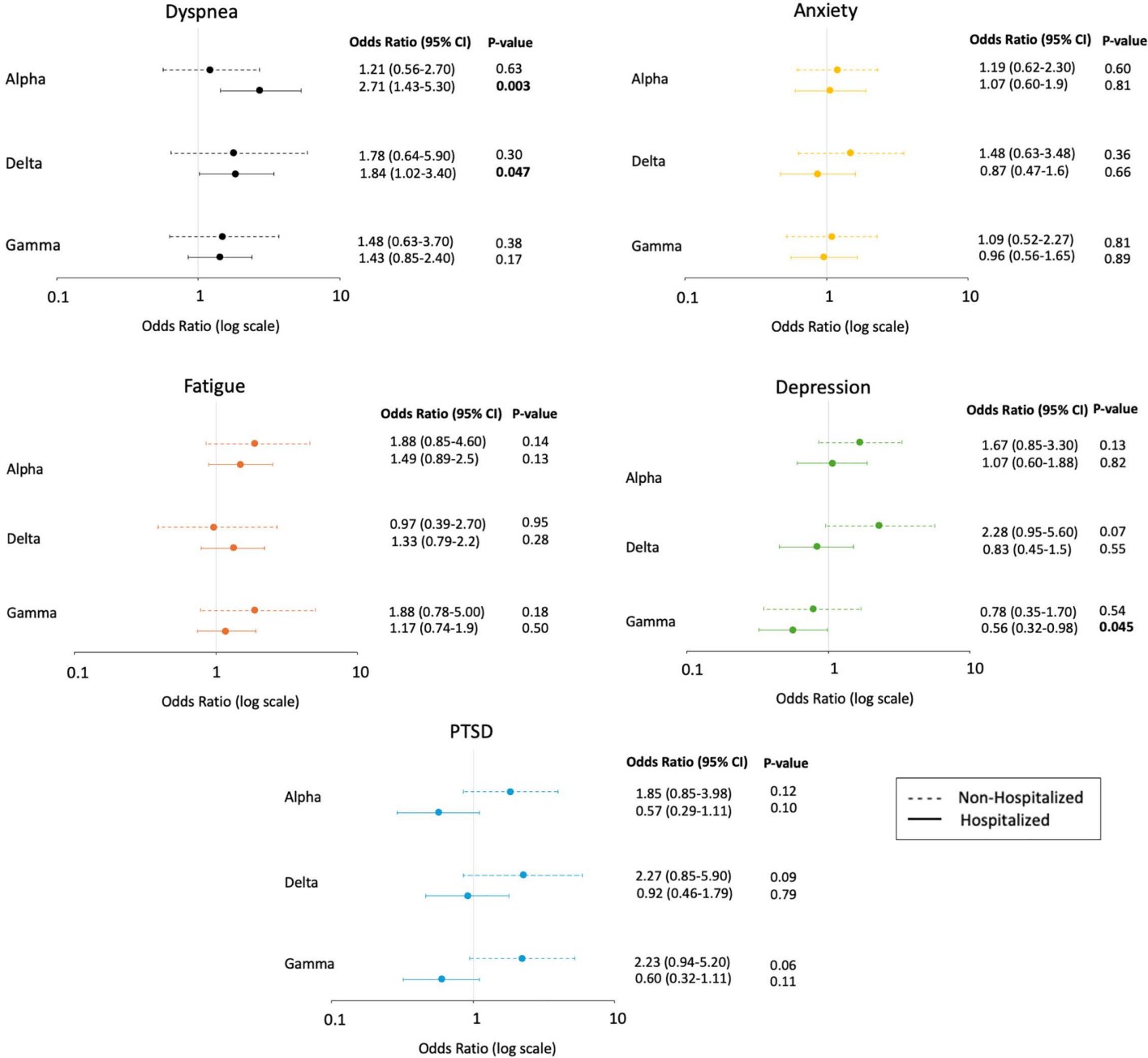

**Fig 3. Association between SARS-CoV-2 variant and abnormal PROM adjusted for age, sex, comorbidities and COVID-19 immunization status.** Reference group: patients infected with the wild-type strain.

Additionally, while the PCRC cohort represents a large cohort of approximately 3800 patients with COVID-19 who have completed validated questionnaires and physiological testing, there is potential referral bias (e.g., care providers in rural locations may not be aware of the PCRCs and those with mild disease may not be referred to the PCRC). While this population may not be fully generalizable, it does allow us to study patients with sustained and severe PCC who would most

**Table 4. Pulmonary function testing based on SARS-CoV-2 variants. The mean±standard deviation of %-predicted values is shown (n=233).**

| SARS-CoV-2 variant | FVC | FEV1 | TLC | DLCO |
|---|---|---|---|---|
| Wild-type (n=152) | 88 ±19 | 88±17 | 89±16 | 86±20 |
| Alpha (n=32) | 80±17 | 82±18 | 81±15 | 71±17 |
| Delta (n=11) | 91±22 | 91±16 | 86±16 | 70±23 |
| Gamma (n=38) | 82±20 | 84±20 | 82±20 | 71±20 |

*Abbreviations:* PFT, pulmonary function test; FVC, forced vital capacity; FEV1, forced expiratory volume in 1 second; TLC, total lung capacity; DLCO, diffusing capacity of lungs for carbon monoxide; SD, standard deviation.

likely require and benefit from further investigations and treatment. Lastly, the variant analysis was limited by small sample sizes, but these exploratory findings can be used to generate subsequent hypotheses and inform future studies.

## Conclusion

Patients who were partially or fully vaccinated did not have increased risk of common PCC symptoms. Infection with the alpha and delta variants was associated with increased likelihood of reporting dyspnea, which may be related to the severity of acute illness and associated impairments in lung function. Further studies on whether different biologic mechanisms could explain the heterogeneity of PCC symptoms between the SARS-CoV-2 variants are need.

## Supporting information

**S1 Table. Baseline PROM questionnaire responses (n=1587).** Abbreviations: UCSD, University of San Diego California. GAD-2, Generalized Anxiety Disorder-2; PHQ-2, Patient Health Questionnaire-2. ◊ The full range of scores for the PROMs are as follows, with higher scores reflecting more symptoms: FSS (sum or mean) (0–63 or 0–7), UCSD (0–120), PTSD (0–5), GAD-2 (0–6), PHQ-2 (0–6). To aid in the interpretation of PROM values, the following normal thresholds have been identified in healthy populations: FSS≤3 (11), PTSD<4 (14 [33]), GAD-2<3 (12 34 ), PHQ-2<3 (13 [34]). * Baseline UCSD scores could not be reported due to missing data. For questionnaires with missing data, UCSD scores were considered abnormal if the values from the completed questions were ≥10/120 (a validated threshold representing the presence of significant dyspnea).
(DOCX)

**S2 Table. SARS-CoV-2 Lineage and Variant Data (n=773).** Participants missing variant data (n=453), Wild type (n=361).
(DOCX)

**S1 Checklist. Human Participants Research Checklist.**
(DOCX)

## Acknowledgments

None.

## Author contributions

**Conceptualization:** Saif-El-Din El-Akkad, Karen C. Tran, Hiten Naik, Naveed Janjua, Hind Sbihi, Christopher Carlsten, James A. Russell, Adeera Levin, Alyson W. Wong.

**Data curation:** Saif-El-Din El-Akkad, Selena Shao, Alyson W. Wong.

**Formal analysis:** Saif-El-Din El-Akkad, Alyson W. Wong.

**Funding acquisition:** Alyson W. Wong.

**Investigation:** Saif-El-Din El-Akkad, Alyson W. Wong.

**Methodology:** Saif-El-Din El-Akkad, Selena Shao, Alyson W. Wong.

**Supervision:** Alyson W. Wong.

**Writing – original draft:** Saif-El-Din El-Akkad, Alyson W. Wong.

**Writing – review & editing:** Saif-El-Din El-Akkad, Karen C. Tran, Hiten Naik, Naveed Janjua, Hind Sbihi, Christopher Carlsten, James A. Russell, Adeera Levin, Alyson W. Wong.

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
