## [Decision Letter · Decision Letter 0]

28 May 2025

Dear Dr. El-Akkad,

Thank you for submitting your manuscript to PLOS ONE. After careful consideration, we feel that it has merit but does not fully meet PLOS ONE’s publication criteria as it currently stands. Therefore, we invite you to submit a revised version of the manuscript that addresses the points raised during the review process.

We look forward to receiving your revised manuscript.

Kind regards,

Dong Keon Yon, MD, FACAAI, FAAAAI

Academic Editor

PLOS ONE

Journal Requirements:

3. You have indicated that data is available from [svitlana.franchuk@ors.ubc.ca].  Please can we ask you to provide us with a general contact email address for the data requests, so readers can request access in perpetuity. If a general email is not available please provide a link to a website where readers can obtain access to data.

4. Thank you for stating the following in your manuscript:

“AWW received funding from the Canadian Lung Association (CLA) / Canadian Institutes of Health Research (CIHR) Respiratory Health Effects of PCC Grant (Funding reference 181075). The funders had no role in study design, data collection and analysis, decision to publish, or preparation of the manuscript. “

“AWW received funding from the Canadian Lung Association (CLA) / Canadian Institutes of Health Research (CIHR) Respiratory Health Effects of PCC Grant (Funding reference 181075). The funders had no role in study design, data collection and analysis, decision to publish, or preparation of the manuscript.”

Additional Editor Comments (if provided):

Thank you for submitting your manuscript. The reviewers and I believe it is of potential value for our readers. However, the reviewers have raised a number of very important issues, and their excellent comments will need to be adequately addressed in a revision before the acceptability of your manuscript for publication in the Journal can be determined. We cannot guarantee that your revised paper will be chosen for publication; this would be solely based on how satisfactorily you have addressed the reviewer comments.

Reviewers' comments:

Reviewer's Responses to Questions

**Comments to the Author**

1. Is the manuscript technically sound, and do the data support the conclusions?

Reviewer #1: Yes

Reviewer #2: Yes

2. Has the statistical analysis been performed appropriately and rigorously?

Reviewer #1: Yes

Reviewer #2: Yes

3. Have the authors made all data underlying the findings in their manuscript fully available?

Reviewer #1: Yes

Reviewer #2: No

4. Is the manuscript presented in an intelligible fashion and written in standard English?

Reviewer #1: Yes

Reviewer #2: Yes

Reviewer #1: Not really novel and somewhat outdated now (alpha, gamma, delta variants will not return) - but could be useful data for some cases, e.g. for those making health insurance claims based on complications related to these older variants, or clinicians with such patients trying to understand their spectrum of illness better for management purposes.

Reviewer #2: This study provides a clinically relevant investigation into the prevalence and determinants of persistent symptoms among COVID-19 survivors, utilizing a cohort derived from a post-COVID care clinic. By examining associations between vaccination status, SARS-CoV-2 variants, and patient-reported outcome measures (PROMs), the study contributes to the growing literature on post-COVID condition (PCC) and offers important insights into modifiable and non-modifiable factors that may influence long-term recovery.

While the authors adjust for several baseline characteristics—including age, sex, and comorbidities—the inclusion of time-dependent covariates would strengthen the analysis. Specifically, stratifying by time since infection or time since last vaccination dose may provide a more nuanced understanding of how temporal dynamics influence post-COVID outcomes.

Although the multivariable models account for several relevant factors, residual confounding remains a concern, particularly with respect to COVID-19 illness severity. Disease severity is likely to influence both the risk of developing post-COVID conditions (PCC) and the probability of being referred to a post-COVID recovery clinic.

The absence of a non-infected comparison group precludes estimating the absolute risk or excess burden of post-COVID conditions. Without a reference population of individuals who were never infected, it is impossible to determine whether the prevalence of abnormal PROMs (e.g., fatigue, dyspnea, anxiety) is elevated beyond what might be expected in the general population or among those with similar comorbidities but without SARS-CoV-2 exposure.

Considering that this is a clinic-based cohort of COVID-19 survivors, there are additional selection biases (referral and survival bias) and generalizability concerns that should be clearly acknowledged in the paper.

**Do you want your identity to be public for this peer review?** For information about this choice, including consent withdrawal, please see our Privacy Policy

Reviewer #1: No

Reviewer #2: No

---

## [Author Response · Author response to Decision Letter 1]

23 Aug 2025

August 21st, 2025

We thank the Editors and Reviewers for their review and thoughtful comments. We hope we have addressed this feedback to their satisfaction.

Sincerely,

Saif El-Akkad, on behalf of the authors

Editor’s Comments:

Response: Thank you for sharing the need for additional style and formatting requirements. Using the provided templates the following changes were made:

• Title page heading changed to size 18 and bolded.

• All author affiliations modified to include department, division, institution, city, province and country.

• Only the email for the corresponding author is now provided.

• Additional text from the title page (word count, figure/table count, keywords) has been removed from title page.

• Abstract has been shortened to 300 words as outlined in submission guidelines.

• Changed headings and subheadings to requested format for level 1 and level 2 headings.

• Changed “figure” to “fig” as requested throughout the manuscript.

• Table 3 and the Figure 3 heading now appear after the paragraph in which they are mentioned.

• Table S1 is now referenced in the manuscript body (page 8): “Baseline PROM scores were also collected and the proportion of participants who reported abnormal scores was calculated (Table S1).”

• An acknowledgements heading is now provided prior to the references, other headings (list of abbreviations, ethics statement, availability of data and materials, conflicts of interest, funding statements and author contributions) have been moved to their respective sections in the manuscript body or removed if not required by PLOS ONE’s formatting requirements.

• Supporting information section now includes both the S1 and S2 tables titles and legends in the requested format.

• Figures and supplemental table files have been renamed according to requested format.

Response: According to the University of British Columbia Clinical Research Ethics Board (UBC CREB), we are prohibited from sharing data publicly as there is potentially sensitive information. Data requests may be sent to the UBC CREB at the following address: Room 210, Research Pavilion - 828 West 10th Ave, Vancouver, BC Canada V5Z1M9

The Data Availability statement in the submission form has been updated to the following: “Data requests may be sent to the UBC CREB at the following address: Room 210, Research Pavilion - 828 West 10th Ave, Vancouver, BC Canada V5Z1M9”

3. You have indicated that data is available from [svitlana.franchuk@ors.ubc.ca]. Please can we ask you to provide us with a general contact email address for the data requests, so readers can request access in perpetuity. If a general email is not available, please provide a link to a website where readers can obtain access to data.

Response: Data requests may be sent to the UBC CREB at the following address: Room 210, Research Pavilion - 828 West 10th Ave, Vancouver, BC Canada V5Z1M9. A general email is not available, but up-to-date contact information can be found on the UBC CREB website: https://researchethics.ubc.ca/clinical-research-ethics/contact-creb

4. Thank you for stating the following in your manuscript:

“AWW received funding from the Canadian Lung Association (CLA) / Canadian Institutes of Health Research (CIHR) Respiratory Health Effects of PCC Grant (Funding reference 181075). The funders had no role in study design, data collection and analysis, decision to publish, or preparation of the manuscript. “

“AWW received funding from the Canadian Lung Association (CLA) / Canadian Institutes of Health Research (CIHR) Respiratory Health Effects of PCC Grant (Funding reference 181075). The funders had no role in study design, data collection and analysis, decision to publish, or preparation of the manuscript.”

Response: The funding statement that was previously provided in the acknowledgements section has been removed. The current funding statement is correct and there are no amendments.

Response: The ethics statement has been removed from the acknowledgement section of the manuscript and is now only mentioned in the methods section as requested.

Reviewer: 1

No additional comments to address.

Reviewer: 2

R2C1. While the authors adjust for several baseline characteristics—including age, sex, and comorbidities—the inclusion of time-dependent covariates would strengthen the analysis. Specifically, stratifying by time since infection or time since last vaccination dose may provide a more nuanced understanding of how temporal dynamics influence post-COVID outcomes.

R2R1: Thank you for this feedback. The median time from infection to completion of PROM questionnaires was approximately 100 days (with interquartile range of 50 days) and similar between the vaccination groups. Time-varying covariates were not included given this relatively short time interval. We have included the median times between infection and questionnaire completion in Table 1.

R2C2. Although the multivariable models account for several relevant factors, residual confounding remains a concern, particularly with respect to COVID-19 illness severity. Disease severity is likely to influence both the risk of developing post-COVID conditions (PCC) and the probability of being referred to a post-COVID recovery clinic.

R2R2: Patients were stratified into those who were hospitalized and were not hospitalized to address disease severity. We have added the following sentence discussing the impact of disease severity as a potential confounder and how we attempted to minimize its impact on the study (discussion paragraph 5).

“We also stratified patients based on whether or not they were hospitalized for COVID-19 as the severity of acute illness may increase the development and severity of PCC (2).”

R2C3. The absence of a non-infected comparison group precludes estimating the absolute risk or excess burden of post-COVID conditions. Without a reference population of individuals who were never infected, it is impossible to determine whether the prevalence of abnormal PROMs (e.g., fatigue, dyspnea, anxiety) is elevated beyond what might be expected in the general population or among those with similar comorbidities but without SARS-CoV-2 exposure.

R2R3: Thank you for raising this valid concern, we agree that this is a key limitation of the study and have added the text below discussing this limitation and how we attempted to minimize its impact on the study (discussion paragraph 5).

“This study is limited by the absence of PROM data prior to COVID-19 infection and lack of a control cohort, which restricts our ability to attribute the presence of abnormal PROMs to PCC. We attempted to focus in on the relationship between COVID-19 infection and abnormal PROMs by adjusting for comorbidities to address the impact that pre-existing conditions and impaired health may have on PROMs.”

We have also added additional text that aids in the interpretation of the reported PROM values and their comparison to normative PROM values reported in the general population (Supplemental Table 1).

“The full range of scores for the PROMs are as follows, with higher scores reflecting more symptoms: FSS (sum or mean) (0-63 or 0-7), UCSD (0-120), PTSD (0-5), GAD-2 (0-6), PHQ-2 (0-6). To aid in the interpretation of PROM values, the following normal thresholds have been identified in healthy populations: FSS ≤ 3 (11), PTSD < 4 (14,33), GAD-2 < 3 (12,34), PHQ-2 < 3 (13,34).”

R2C4. Considering that this is a clinic-based cohort of COVID-19 survivors, there are additional selection biases (referral and survival bias) and generalizability concerns that should be clearly acknowledged in the paper.

R2R4: We thank reviewer 2 for raising this suggestion. Concerns around referral bias and the generalizability of this cohort have been addressed in the text below (discussion paragraph 5).

“Additionally, while the PCRC cohort represents a large cohort of approximately 3800 patients with COVID-19 who have completed validated questionnaires and physiological testing, there is potential referral bias (e.g., care providers in rural locations may not be aware of the PCRCs and those with mild disease may not be referred to the PCRC). While this population may not be fully generalizable, it does allow us to study patients with sustained and severe PCC who would most likely require and benefit from further investigations and treatment.”

---

## [Decision Letter · Decision Letter 1]

9 Sep 2025

Dear Dr. El-Akkad,

Thank you for submitting your manuscript to PLOS ONE. After careful consideration, we feel that it has merit but does not fully meet PLOS ONE’s publication criteria as it currently stands. Therefore, we invite you to submit a revised version of the manuscript that addresses the points raised during the review process.

We look forward to receiving your revised manuscript.

Kind regards,

Dong Keon Yon, MD, FACAAI, FAAAAI

Academic Editor

PLOS ONE

Journal Requirements:

Additional Editor Comments:

# COVID infection. -> SARS-CoV-2 infection

# The BCCDC PHL continuously monitors for VOCs, variants of interest (VOIs), and variants under monitoring (VUMs). ?? -> Please describe definition of VOC duration.

# COVID-19 infection -> SARS-CoV-2 infection

# COVID vaccination -> COVID-19 vaccination

# demonstrated -> found

Reviewers' comments:

Reviewer's Responses to Questions

**Comments to the Author**

Reviewer #2: All comments have been addressed

2. Is the manuscript technically sound, and do the data support the conclusions?

Reviewer #2: Yes

3. Has the statistical analysis been performed appropriately and rigorously?

Reviewer #2: Yes

4. Have the authors made all data underlying the findings in their manuscript fully available?

Reviewer #2: Yes

5. Is the manuscript presented in an intelligible fashion and written in standard English?

Reviewer #2: Yes

Reviewer #2: (No Response)

**Do you want your identity to be public for this peer review?** For information about this choice, including consent withdrawal, please see our Privacy Policy

Reviewer #2: No

---

## [Author Response · Author response to Decision Letter 2]

29 Oct 2025

October 11th, 2025

We thank the Editors and Reviewers for their review and thoughtful comments. We hope we have addressed this feedback to their satisfaction.

Sincerely,

Dr. Saif El-Akkad, on behalf of the authors

Editor’s Comments:

1. The following additional editor comments were made:

a. COVID infection -> SARS-CoV-2 infection

b. COVID-19 infection -> SARS-CoV-2 infection

c. COVID vaccination -> COVID-19 vaccination

d. Demonstrated -> found

Response: Thank you for sharing the need for additional style and formatting requirements. The following changes were made:

• COVID infection has been changed to SARS-CoV-2 infection.

• COVID-19 infection has been changed to SARS-CoV-2 infection.

• COVID vaccination has been changed to COVID-19 vaccination.

• Demonstrated has been changed to found.

2. Please describe definition of variant of concern (VOC) duration.

Response: Text has been added to clarify the time periods during which variants of concern circulated and the period during which they were studied (methods paragraph 4):

VOCs circulated in British Columbia during the following approximate time periods: alpha (January – June 2021), gamma (February – July 2021), delta (May 2021 – December 2021) and omicron (December 2021 onwards) (17). Data were limited to these four dominant VOCs based on data availability and categorization used by the BCCDC Public Health Laboratory (PHL) during the study duration (March 2020 to October 2022).

Please let us know if we did not satisfactorily describe the VOC duration with our response above. We would be happy to re-address this comment with further clarification on what VOC duration refers to.

Reviewer Comments:

No additional comments to address.

---

## [Editor Report · Decision Letter 2]

2 Nov 2025

Association between COVID-19 vaccination, SARS-CoV-2 variants, and post COVID-19 condition: a cross-sectional study

PONE-D-25-00682R2

Dear Dr. El-Akkad,

We’re pleased to inform you that your manuscript has been judged scientifically suitable for publication and will be formally accepted for publication once it meets all outstanding technical requirements.

Kind regards,

Dong Keon Yon, MD, FACAAI, FAAAAI

Academic Editor

PLOS ONE

Additional Editor Comments (optional):

This is an excellent paper.
---

## [Editor Report · Acceptance letter]

PONE-D-25-00682R2

PLOS ONE

Dear Dr. El-Akkad,

I'm pleased to inform you that your manuscript has been deemed suitable for publication in PLOS ONE. Congratulations! Your manuscript is now being handed over to our production team.

Kind regards,

on behalf of

Dr. Dong Keon Yon

Academic Editor

PLOS ONE